# STARCall integrates image stitching, alignment, and read calling to enable scalable analysis of *in situ* sequencing data

**Nicholas J. Bradley**[1], **Sriram Pendyala**[1,2], **Katie Partington**[1,3], **Douglas M. Fowler**[1,3,4]*

**1** Department of Genome Sciences, University of Washington, Seattle, Washington, United States of America, **2** Medical Scientist Training Program, University of Washington, Seattle, Washington, United States of America, **3** Department of Bioengineering, University of Washington, Seattle, Washington, United States of America, **4** Brotman Baty Institute for Precision Medicine, Seattle, Washington, United States of America

* dfowler@uw.edu

## Abstract

Fluorescent *in situ* sequencing involves imaging-based sequencing by synthesis in intact cells or tissues to reveal target nucleotide sequences inside each cell. Often, the target sequences are barcodes that indicate a perturbation (e.g., CRISPR guide or genetic variant) delivered to the cell. However, processing *in situ* sequencing data presents a considerable challenge, requiring stitching and aligning tens of thousands of images with millions of cells, detecting small amplicon colonies across sequencing cycles, and calling reads. To address these challenges, we introduce STARCall: STitching, Alignment and Read Calling for *in situ* sequencing, a software package that analyzes raw *in situ* sequencing images to produce a genotype-to-phenotype mapping for each cell. STARCall improves upon previous solutions by combining stitching and alignment of images into a single step that minimizes both inter-cycle and intra-cycle alignment error. STARCall also improves detection and extraction of sequencing reads, incorporating filters and normalization to combat background fluorophore signal. We compare STARCall to other methods using a diverse set of images that include commonly encountered imaging problems such as variable intensity across channels and cycles and high levels of background. Specifically, this comprises ~250,000 images from a pooled screen of ~3,500 barcoded LMNA variants expressed in U2OS cells and ~1,200 barcoded PTEN variants in induced pluripotent stem cells (iPSC) and iPSC-derived neurons. Overall, STARCall aligned more than 50% of tiles with <1 pixel residual misalignment on all nine image sets, outperforming alternative packages by 14–35%. STARCall also yielded an 8–40% increase in genotyped cells due to improved filtering and normalization methods that address background fluorescence. STARCall can call tools like CellPose to segment cells and CellProfiler to compute cell features from the phenotyping images. STARcall is

**Data availability statement:** STARCall is freely available under the MIT licence. STARCall can be found at https://github.com/FowlerLab/starcall-workflow. Instructions on installation and running STARCall, as well as a small example dataset are available at this repository. Documentation for the two Python libraries is available at https://fowlerlab.github.io/starcall-docs/constitch.html for ConStitch and https://fowlerlab.github.io/starcall-docs/starcall.html for STARCall. The image sets used to measure stitching and read calling performance are available for download through this link: https://g-e16ca0.7067fc.8443.data.globus.org/nobackup/public/starcall-testing-data-sets-20250925/index.html. In addition, data tables of values used to create the figures of this paper and values behind all summary statistics are hosted on Zenodo at https://doi.org/10.5281/zenodo.19027452.

**Funding:** This work was supported by the NIH (RM1HG010461 to DMF) and the Chan Zuckerberg Initiative (CZIF2024-010284 to DMF). The funders had no role in study design, data collection and analysis, decision to publish, or preparation of the manuscript.

open-source and freely available, providing a robust solution for the analysis of *in situ* sequencing data.

---

## Author summary

Short regions of RNA or DNA can be sequenced inside intact cells or tissues (i.e., *in situ*) by combining a microscope and sequencing by DNA synthesis. Multiple cycles of sequencing are performed, in which incorporation of a single fluorescently labeled nucleotide is imaged, and the corresponding base detected. Recently, *in situ* sequencing has proved useful in optical pooled screens, where a library of perturbations such as CRISPR-mediated gene knockouts or genetic variants is introduced into cells, and *in situ* sequencing is used to reveal the specific perturbation in each cell. However, processing *in situ* sequencing data presents a considerable challenge, requiring stitching and aligning tens of thousands of images, detecting small amplicon colonies across sequencing cycles, and calling reads. To address these challenges, we introduce STARCall: STitching, Alignment and Read Calling for *in situ* sequencing. STARCall uses a stitching algorithm that minimizes both inter-cycle and intra-cycle misalignment, and improved filters and normalization for base calling. When applied to a set of 9 *in situ* sequencing image sets, STARCall yielded an 8–40% increase in genotyped cells. STARCall is open-source and applicable to a variety of experiments, providing a robust pipeline for *in situ* sequencing data.

## Introduction

Fluorescent *in situ* sequencing [1–3] involves imaging-based sequencing by synthesis in intact cells or tissues to reveal nucleotide sequences of interest inside each cell. PCR or rolling circle amplification (RCA) colonies comprising many copies of target sequences are created and then cycles of sequencing are conducted. In each cycle, polymerase-mediated addition of reversibly terminated, fluorescently labeled nucleotides is followed by imaging. Then, the terminating dye is cleaved and the next cycle conducted. Recently, *in situ* sequencing has found diverse applications, especially for optical pooled screening [4]. Here, perturbations such as CRISPR-mediated gene knockouts are read out via *in situ* sequencing of short barcodes to reveal the identity of the perturbation in each cell [5,6]. We also recently developed Variant *In Situ* sequencing (VIS-seq), where a library of protein variants is introduced to cells and sequencing of a short barcode is used to identify the variant of each cell [7].

*In situ* sequencing involves generating a large set of imaging data comprising phenotyping images, taken to characterize molecular and cellular features of interest, and images of every cell in each cycle of sequencing. The number of cycles is determined by the length of the target sequence, typically 8–12 bases for barcodes in optical pooled screening. A cycle of sequencing comprises images

from each of four channels corresponding to the four nucleobases and a fiducial channel to enable alignment across cycles (Fig 1A). A typical optical pooled screening imaging set consists of ~25,000 sequencing images encompassing millions of cells [4,8].

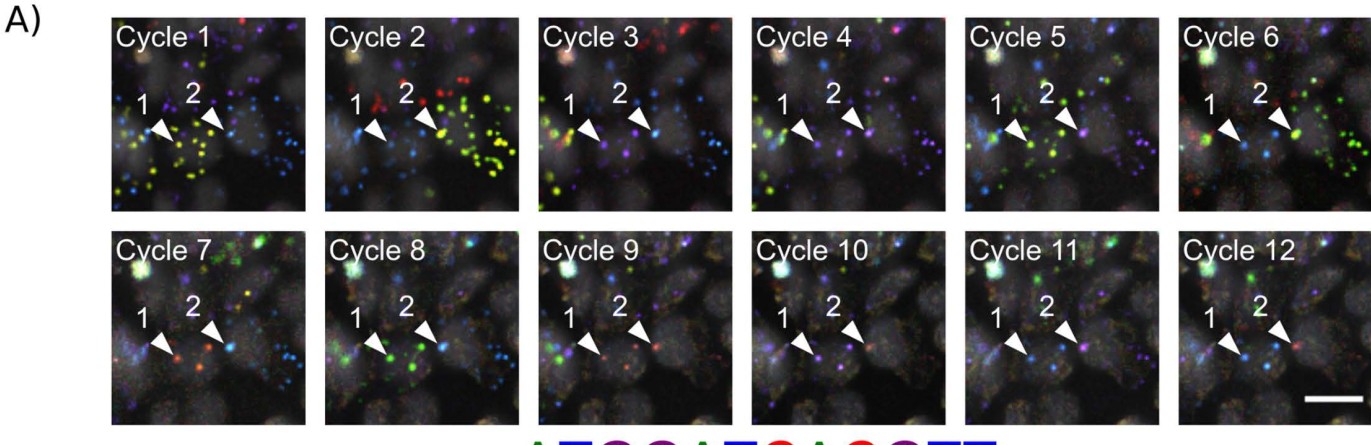

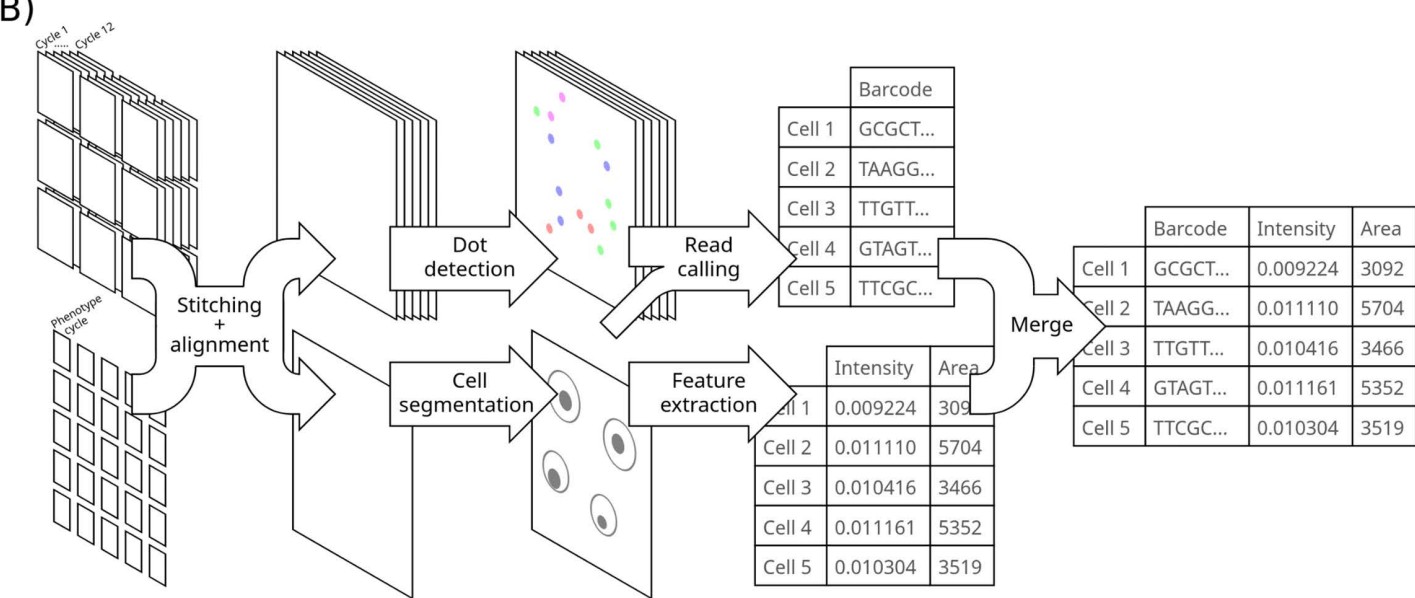

**Fig 1.** *In situ* **sequencing involves image stitching and alignment, dot detection, genotyping and phenotyping of cells. A)** Barcodes introduced into cells are read out with 12 cycles of sequencing-by-synthesis. The amplicon colonies that correspond to two barcodes in different cells are highlighted in each cycle, and the sequences shown below. A 10 micron scale bar is included in the bottom right image. **B)** Stitching and alignment must be performed on both sequencing and phenotyping images. Then, amplicon colonies (dots) are detected and cells are segmented. Reads are generated from amplicon colonies and assigned to cells, and various features are calculated for each cell to represent phenotypes. Two example features, area and intensity, are shown.

Images must first be stitched together, because the microscope collects many fields of view for each well (Fig 1B). Images must also be aligned across sequencing cycles because the microscope cannot revisit each field of view in each cycle with perfect accuracy, and the well plate is typically removed between cycles to add reagents, meaning that images are not in a common coordinate space (Figs 1A and 2A). Moreover, phenotype images may be acquired using a different objective or an entirely different microscope yet must also be stitched and aligned. After stitching and alignment, reads are called by detecting amplicon colonies, which appear as small, bright dots in cells. The channel in which the dot is fluorescing in each cycle is identified and dots are assigned to cells. Often, consensus reads in each cell must then be determined and matched to a barcode lookup table. Thus, image stitching, alignment, amplicon colony detection and read calling comprise the major computational challenges for the analysis of *in situ* sequencing data.

Previously, ASHLAR [9] (Alignment by Simultaneous Harmonization of Layer/Adjacency Registration) has been used to stitch and align multi-cycle image sets. ASHLAR employs the MIST [10] (Microscopy Image Stitching Tool) algorithm to stitch the first cycle, then aligns each successive cycle onto the first [8]. Alternatively, the well plate can be manually aligned on the microscope while imaging, ensuring each field of view is in the same location across all cycles and removing the need for computational alignment. Following stitching and alignment, the location of amplicon colonies has been detected by calculating the per-pixel standard deviation across cycles, which amplifies pixels that are changing frequently, then finding local maxima in the resulting image [4]. Cell segmentation, required to map reads to cells, has been accomplished using deep learning methods such as Cellpose [11] or Stardist [12]. Finally, specific features describing the intensity, distribution or texture of stains in the phenotype images are often extracted with CellProfiler [13].

The stitching and alignment requirements for *in situ* sequencing image sets, including alignment of images collected with different objectives, near perfect alignment between cycles, and stitching of regions with sparse features, are solvable with current stitching and alignment algorithms. However, the number of requirements means that most existing implementations cannot perform all tasks needed. For example, ASHLAR is the current state-of-the-art package for cycle-to-cycle alignment, but it does not support alignment between images collected with different objectives. Background and intensity differences across cycles and channels complicate read calling, and the miscalling of a single base can corrupt entire sequences in these costly experiments. Additionally, no unified pipeline exists that handles the stitching, alignment, genotyping, and phenotyping of cells.

To resolve these challenges, we present STARCall (STitching Alignment and Read Calling for *in situ* sequencing), a robust software pipeline that performs all the steps needed to call short reads and extract phenotype features from raw image sets. In STARCall, images are first stitched and aligned with an algorithm that minimizes both inter-cycle and intra-cycle alignment error. This resulted in more than 50% of tiles with <1 pixel residual misalignment on all nine image sets, outperforming ASHLAR by 14–35%. From the stitched, aligned sequencing images, STARCall identifies amplicon colonies and calls reads using a Gaussian filter and the Laplacian of Gaussian blob detection algorithm [14], which detected amplicon colonies better than previous methods in a variety of image conditions. Together, these improvements yielded a 8–40% increase in the number of cells genotyped across a range of barcode-based *in situ* sequencing experiments. STARCall can also call tools such as CellPose [11] to segment cells, BaSiC [15] to perform illumination correction, and CellProfiler [13] to compute cell features from the phenotyping images, accepting user-created CellProfiler pipelines to customize the features calculated. STARCall is freely available, implemented in Python 3.10, and provides both a ready-to-use Snakemake [16] pipeline as well as a set of customizable Python libraries, independently enabling image stitching, alignment, and read calling.

## Results

### Stitching and alignment of multi-cycle imaging datasets

Stitching and alignment of *in situ* sequencing image sets is critical as all downstream processing steps rely on having accurately aligned images. Because the relevant features for *in situ* sequencing images are small (<3 pixels) RCA colony-derived dots, stitching and alignment must be near pixel-perfect. Misalignment of a single cycle could result in the

 

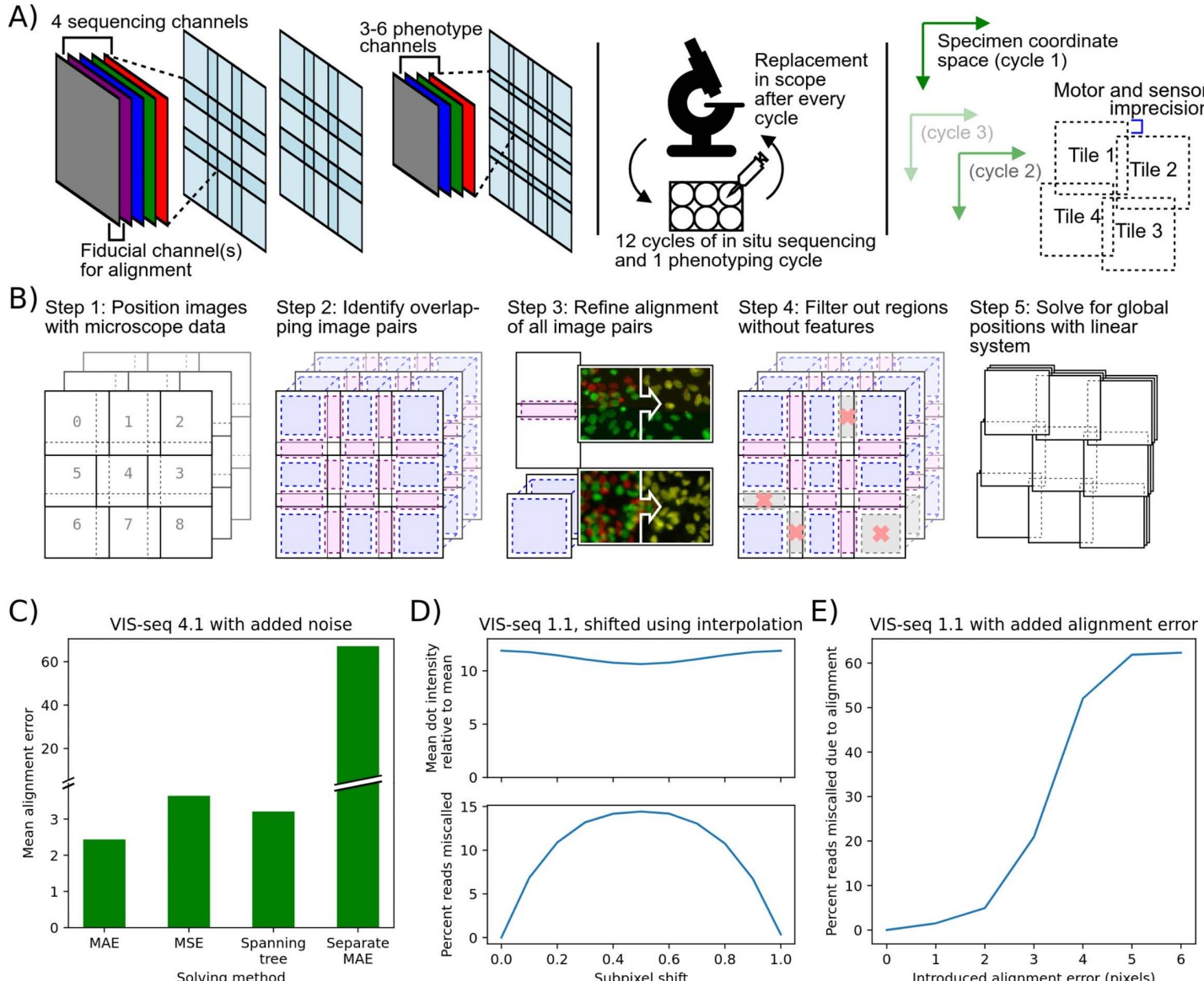

**Fig 2. ConStitch performs stitching and alignment jointly on multi-cycle image sets. A)** Imaging procedure for *in situ* sequencing. Each sequencing cycle contains four channels corresponding to each nucleotide, and a fiducial channel to help with alignment. Positional errors are introduced into the collected images in two ways: intra-cycle, due to inaccuracies of microscope stage motor and sensors, and inter-cycle, due to removal and replacement of the well plate from the microscope. **B)** Procedure for stitching and aligning multi-cycle image sets. Overlapping pairs of images are aligned to each other, regions with no features are excluded from the alignment process, then the system of equations containing all pairwise alignments is solved to find global positions for each image. **C)** Performance of different solving methods with high levels of noise: Mean absolute error solver (default), mean squared error solver, spanning tree solver, and an individual cycle solver where each cycle is stitched independently with the MAE solver, then all stitched cycles are aligned with a single translation. Performance is tested on image set 4.1. To better show performance differences in suboptimal conditions, Gaussian noise of sigma = 100 was added to the pixel intensities of the image. **D)** Ratio of mean dot intensity to mean pixel intensity (top) and percent of reads miscalled (bottom) when sequencing images are shifted by a fractional pixel offset using spline interpolation. **E)** Percent of reads miscalled when alignment error is introduced into a random cycle.

miscalling of all bases in that cycle, with the consequent corruption of all reads. Two state-of-the-art software packages, MIST [10] and ASHLAR [9], are often used for microscopy image stitching and alignment. MIST stitches microscopy images, but does not support alignment across cycles. ASHLAR supports both stitching and alignment across cycles, but it lacks key features needed for processing *in situ* sequencing image sets, namely stitching of cycles taken with different objectives. To address these missing features, we developed ConStitch, a package that is a core part of STARCall and that can also be used independently.

Like MIST and ASHLAR, ConStitch operates on an adjacency graph of all images, where each image corresponds to a node and edges connect all pairs of overlapping images. ConStitch extended this approach to accomplish cycle-to-cycle alignment by including images from all cycles in a single adjacency graph. Two types of edges are added: intra-cycle edges connect adjacent tiles that are in the same cycle, and inter-cycle edges connect tiles in the same position but in different cycles. Thus an adjacency graph containing all cycles is constructed, which allows for joint optimization of stitching and alignment. The adjacency graph is constructed using positional information from the microscope, or a user-defined scanning pattern. (Fig 2B step 1). Since all cycles are included in the adjacency graph, images receive an initial x and y position as well as a cycle number. Each edge is annotated with the x and y offset that relates the two images, which we refer to as a positional constraint (Fig 2B step 2). Phase cross correlation [17,18] is applied to each pair of overlapping images, refining the initial constraint by maximizing the cross correlation between the images (Fig 2B step 3). If the two images have differing scale factors, e.g., they were taken with different objectives, the image with the smaller pixel size is scaled up to match the resolution of the other image before phase cross correlation is applied. The pixel size of images is typically included in the metadata recorded by the microscope, and is extracted by ConStitch. Next, the alignment for each edge is scored with the zero normalized cross correlation (ZNCC), calculated as the dot product of the overlapping pixel intensities, normalized to a unit vector with mean zero. Poorly aligned image pairs are filtered out by comparing the ZNCC score for each image pair to the 95th percentile ZNCC of a random set of non-overlapping image pairs (Fig 2B step 4). A linear model is trained on the remaining constraints to estimate physical parameters of the microscope such as the travel between tiles and the camera angle. We use these parameters to estimate constraints for the poorly aligned image pairs that were removed during filtering.

To find the global positions of all images (Fig 2B step 5), we first used a maximum spanning tree like MIST and ASHLAR, but found it did not provide adequate alignment in suboptimal imaging conditions, when there may be many erroneous constraints. An alternative approach previously used to stitch 3D image sets is to minimize the mean squared error of all pairwise constraints, searching over the space of all possible global positions [19]. To find the set of positions that minimizes the error, we represent the constraint graph as an overconstrained linear system of equations, where each constraint contains equations relating the x and y positions of the two images it contains. This method benefits from existing solvers of systems of equations and various loss functions. We tested two methods of solving, minimizing either mean squared error (MSE) or mean absolute error (MAE). In addition we tested a solving method that uses the MAE solver on each cycle independently, then aligns the stitched cycles to each other. We included this to confirm that joint optimization of all cycles is vital for achieving low alignment error. The spanning tree, MSE solver, and MAE solver methods had similar performance on images that align well, but the MAE solver performed better in non-optimal imaging conditions. We believe this is due to the MAE solver being robust to outliers, which enables it to handle erroneous constraints that are still present after filtering. Specifically, the MAE solver had the lowest alignment error on a set of *in situ* sequencing images to which we added noise (Table 1 and Fig 2C). Thus, ConStitch employs joint optimization of all cycles with the MAE solver to find the optimal positions for each image.

To merge all images together into a stitched image, the positions for each image must be integers. Simply rounding the optimal, floating point positions found by the MAE solver results in integer positions that are no longer

**Table 1. Image sets used to test the performance of STARCall compared to MIST, ASHLAR and the Feldman et al pipeline.**

| Image Set | Specimen | Fiducial Stains | Objective | # Images | # Cycles | % Overlap | Used to compare | Notes |
|-----------|----------|-----------------|-----------|----------|----------|-----------|-----------------|-------|
| MIST | Stem cell colonies | GFP | 10X | 5318 | 5 (rounds) | 10% | Stitching | Imaged after two days of growth |
| ASHLAR | Slice of human colon | Hoechst 33342 | 20X | 1218 | 2 | 3% | Stitching | |
| Feldman | A549 cells | DAPI | 10X | 5544 | 9 | 5% | Both | |
| VIS-seq 1.1 | U2OS cells | GFP, DAPI | 10X, 20X | 5868 | 12 | 15% | Both | |
| VIS-seq 1.2 | U2OS cells | GFP, DAPI | 10X, 20X | 5868 | 12 | 15% | Both | |
| VIS-seq 2.1 | U2OS cells | GFP, DAPI | 10X, 20X | 5868 | 12 | 15% | Both | |
| VIS-seq 2.2 | U2OS cells | GFP, DAPI | 10X, 20X | 5868 | 12 | 15% | Both | |
| VIS-seq 3.1 | iPS cells | GFP, DAPI | 10X, 20X | 3912 | 8 | 15% | Both | Double barcode |
| VIS-seq 3.2 | iPS cells | GFP, DAPI | 10X, 20X | 3912 | 8 | 15% | Both | Double barcode |
| VIS-seq 4.1 | iPSC differentiated neurons | GFP, DAPI | 10X, 20X | 3912 | 8 | 15% | Both | Double barcode |
| VIS-seq 4.2 | iPSC differentiated neurons | GFP, DAPI | 10X, 20X | 3912 | 8 | 15% | Both | Double barcode |

optimal, introducing ~1 pixel of alignment error across the image set. Instead we constrained the solution to integer values using integer linear programming which ensured an optimal solution, but the computational costs were highly variable, making this approach impossible on some image sets. ASHLAR handles this rounding problem by interpolating each image to account for the fractional component of its position, resulting in near zero alignment error. We suspected this interpolation could affect the detection of amplicon colonies as small features are not always preserved during interpolation, and after testing found that images shifted by 0.5 pixels reduced the intensity of dots by ~10% relative to the mean pixel intensity, and caused 15% of reads to be miscalled (Table 1 and Fig 2D). Conversely, the small 1 pixel misalignment introduced by our rounding procedure causes only 1.5% of reads to be miscalled (Fig 2E). Thus, either through integer linear programming or rounding we obtain integer positions for each image, which allows us to embed all images in a common coordinate space, merge shared regions by taking the mean of every overlapping pixel, and create the final composite set of aligned images.

## Global stitching and alignment improves performance

We evaluated the performance of ConStitch compared to MIST [10] and ASHLAR [9] on a variety of datasets. The image dataset used in the original MIST publication consists of a pair of single cycle imaging sets of a well of GFP labeled stem cells. The first set of images was acquired using a scanning pattern with 10% overlap between each adjacent image. The second set of images was acquired with each colony of stem cells manually centered and fully contained in the field of view, while additionally recording the location of the colony with the microscope stage position. We downloaded this evaluation image set as well as full well images that were stitched with MIST, and used ConStitch to stitch additional full well images. We segmented the full well images from ConStitch and MIST to identify colonies using the threshold described in Chalfoun, J. et al [10]. Stitching was scored by first matching stem cell colonies in the full well images to the closest colony in the evaluation set. The matched colonies were then compared based on the total area of the segmented colony and the colony position recorded by the microscope. The distributions of area error and positional error were similar, with ConStitch providing slightly higher area error and slightly lower positional error (Fig 3A and 3B). Thus, ConStitch and MIST performed similarly for stitching of single-cycle image sets.

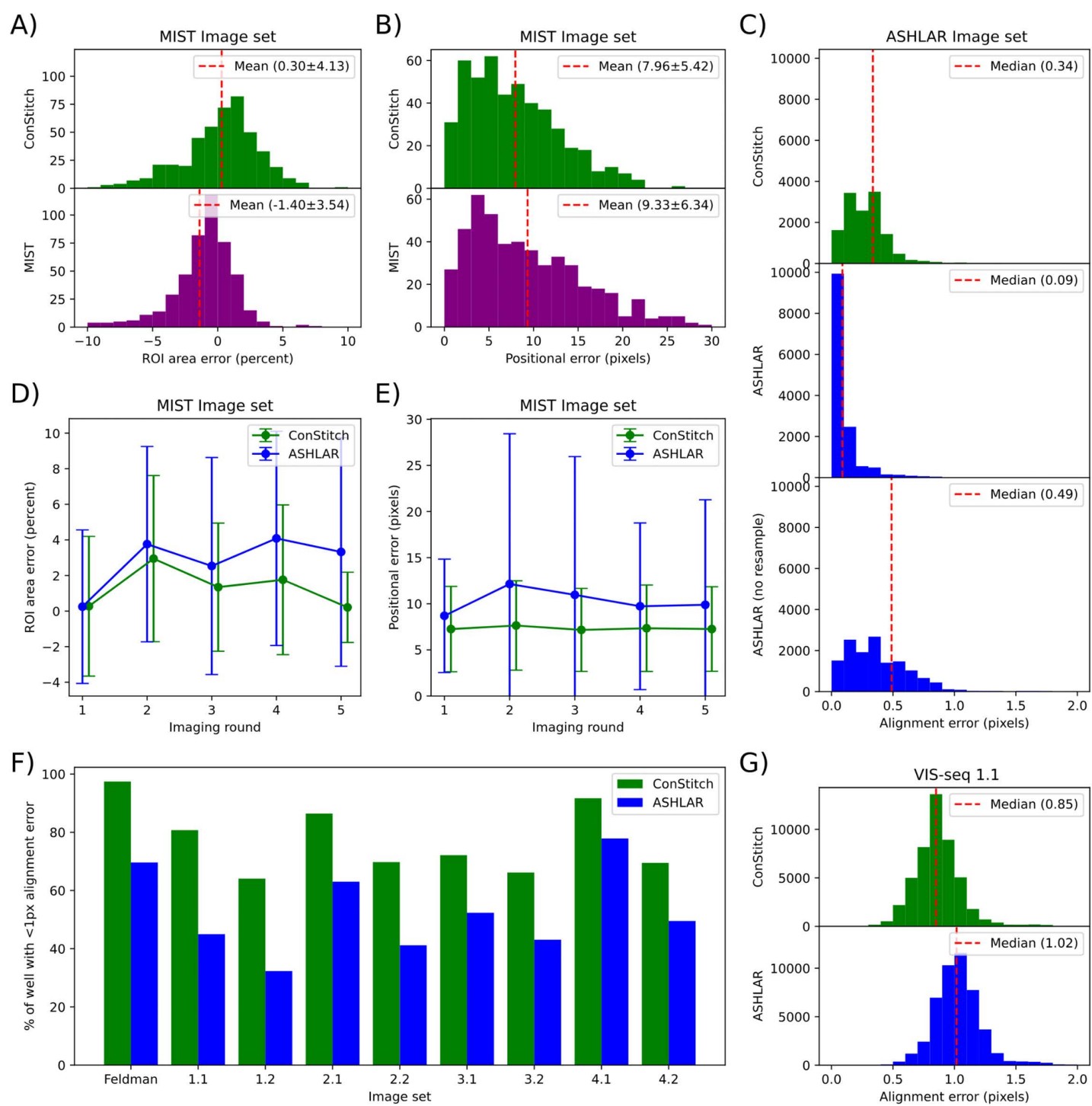

**Fig 3. ConStitch provides comparable stitching performance to MIST and superior alignment performance to ASHLAR. A-B)** Stitching accuracy of MIST and ConStitch using the MIST evaluation framework, measuring differences in colony area (A) and colony position (B) compared to ground truth images. **C)** Alignment error of ASHLAR and ConStitch on ASHLAR evaluation dataset derived by breaking stitched and aligned images into 200 pixel by 200 pixel blocks, the alignment of which was evaluated between the two cycles with phase correlation. Histograms show magnitudes of alignment vectors for blocks in the resulting optical flow field. **D-E)** Stitching performance of ConStitch and ASHLAR on all imaging rounds of the MIST image set when aligned together. Area error and positional error are reported across the five imaging rounds. **F)** Percent area of well with less than 1 pixel of alignment error. Alignment error was measured as in (C), applied to each pair of cycles, taking the maximum error at each location. **G)** Histogram of alignment error on image set 1.1, calculated as in (C).

To further evaluate the performance of ConStitch we applied it to the ASHLAR evaluation dataset [9], which comprises two cycles of imaging from a cyclic immunofluorescence [20] experiment performed on a 5 µm thick section of a human colon tissue sample, with nuclei stained using Hoechst 33342 in both cycles. We measured alignment performance by dividing the stitched images into 200 by 200 pixel regions, then applying phase cross correlation to each region to find the misalignment between cycles. This metric does not quantify misalignment relative to any ground truth, instead only measuring the consistency of the alignment against itself. Nonetheless, this metric is still valuable as we already confirmed ConStitch provides comparable stitching performance to MIST and ASHLAR, and we showed how small misalignments can be detrimental to read calling (Figs 2E, 3A, and 3B). ConStitch yielded a higher residual misalignment than ASHLAR using subpixel interpolation, and lower misalignment than ASHLAR without interpolation (Fig 3C). We also tested the alignment of ASHLAR and ConStitch using the MIST image set mentioned previously. This image set contains multiple rounds of imaging done with different levels of overlap (10% overlap to 50%). We used ASHLAR and ConStitch to align these rounds onto each other, and measured the error in colony area and position for each round (Fig 3D and 3E). ASHLAR and ConStitch perform similarly in the first round, but in later rounds ASHLAR's performance degrades. We believe this is due to ASHLAR only stitching the first round, then aligning all tiles in subsequent rounds to the first round, meaning that later rounds are only indirectly stitched. ConStitch calculates the alignment of every overlapping tile pair and finds the optimal solution, meaning that the stitching of every round is optimal. In fact, the positional error achieved on the first round (7.26±4.62) is lower than the error achieved when it was stitched independently (7.96±5.42), showing that ConStitch uses information from all cycles to optimize stitching (Fig 3B and 3E).

We also compared the performance of ConStitch and ASHLAR [9] on four VIS-seq datasets we generated [7] and a CRISPR screen from Feldman et al. [4] (Table 1). Each of the VIS-seq experiments has two image sets representing replicate experiments. VIS-seq image sets 1 and 2 involve lamin A protein variants expressed in the U2OS human cancer cell line. VIS-seq images sets 3 and 4 involve PTEN protein variants expressed in iPS cells and iPS cell-derived neurons. Compared to the MIST and ASHLAR evaluation image sets we analyzed, the VIS-seq image sets contain more images, more cycles and a larger degree of overlap. These image sets also contain more than two cycles; to summarize the alignment at each location we take the maximum misalignment between all pairs of cycles. As shown previously, a misalignment of 1 pixel has a minimal effect on read calling performance (Fig 2E). Thus, to represent the quality of alignment for an image set we measure the "alignment percentage", which is the percent area of the stitched image that has a misalignment of less than 1 pixel. Due to the damaging effects of interpolation on *in situ* sequencing image sets shown before, we disabled the interpolation feature of ASHLAR before stitching (Fig 2D).

ConStitch had a 10–30% higher alignment percentage than ASHLAR on all image sets (Fig 3F). The largest discrepancy was on image set 1.1, with ConStitch yielding 81% of pixels aligned with an error less than 1 pixel compared to 45% for ASHLAR. This difference in performance is also reflected in the respective distributions of misalignment (Fig 3G). Thus, ConStitch delivers small but critical improvements in alignment accuracy and also provides features such as the ability to stitch images from different objectives or microscopes, making ConStitch better suited for stitching *in situ* sequencing image sets compared to ASHLAR.

## New methods of background filtering and normalization improve read calling results

In the stitched images, amplicon colonies give rise to dots that fluoresce in one of four channels in every cycle, each channel corresponding to a different nucleobase (Fig 4A). First, the signal from each dot in a single cycle is analyzed to call the base, then base calls are collected across cycles to form the read associated with the amplicon colony. Reads must then be assigned to the correct cell, and one or more consensus sequences determined from the reads assigned to the cell. STARCall includes a pipeline to call bases and cell consensus read sequences, inspired by Feldman et al. [4]

To develop STARCall's base and read calling algorithm, we leveraged the distinctive optical features of amplicon colonies. The colonies appear as small circular dots in each cycle, bright in only one of the channels. Across cycles, signal appears in different channels for each dot, reflecting the different bases at each position of the sequence. Detection is complicated by a background level of fluorophore that builds up and varies over cycles, primarily in the C and A channels (Fig 4B and 4C). Additionally, the brightness of the amplicon colonies decreases over cycles, dropping off rapidly in the initial cycles and less rapidly in subsequent cycles. The

PLOS Computational Biology

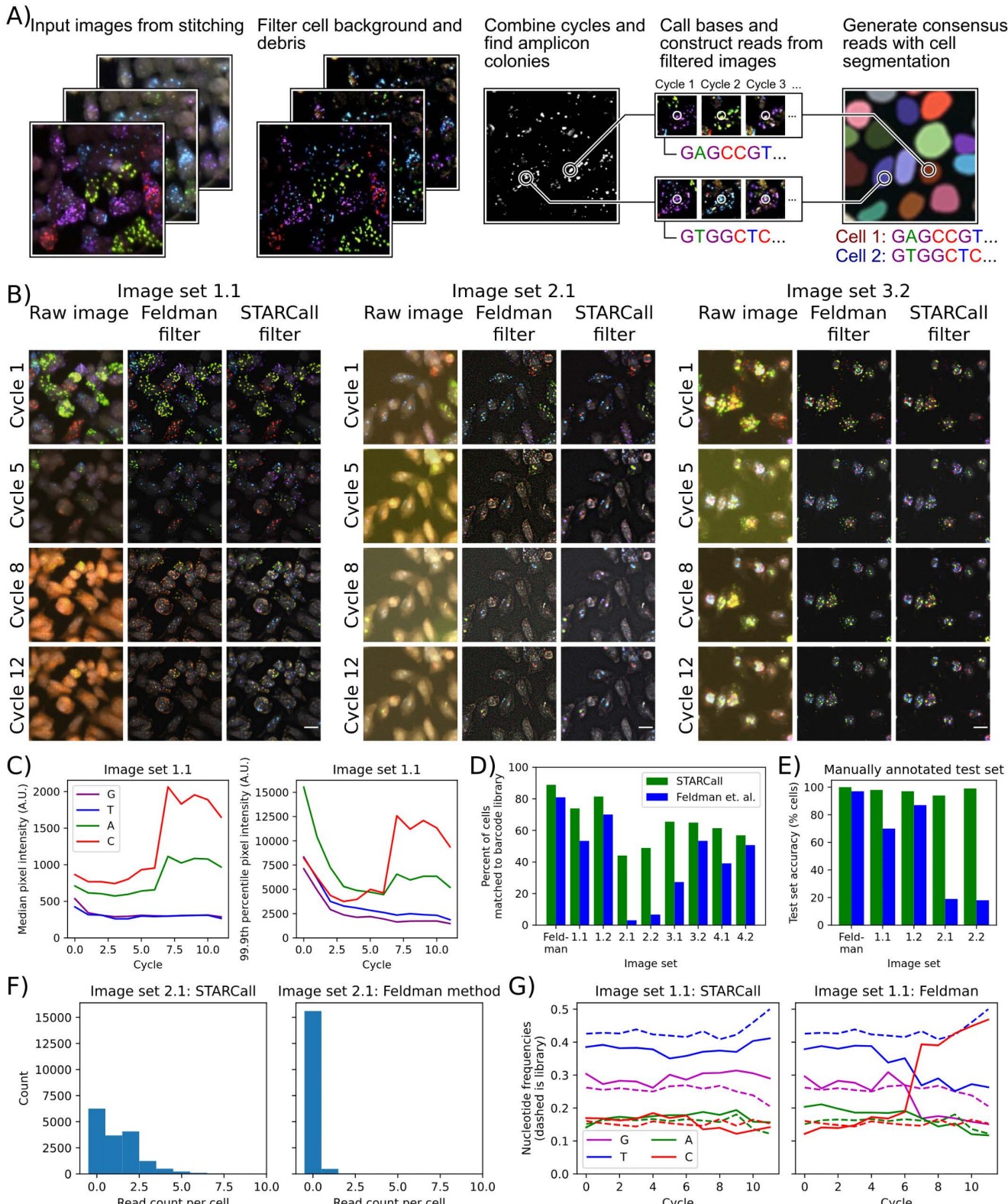

**Fig 4. STARCall filters out background and corrects for differences in intensity when calling reads. A)** Key steps for read calling by STARCall, including background filtering, amplicon colony detection, base calling and read assembly. **B)** Sequencing images before and after the Laplacian of

Gaussian filter from Feldman et al. and the Gaussian filter and normalization from STARCall. Amplicon colonies in each cell appear as small bright dots in the channel corresponding to their nucleobase. Violet = G, Blue = T, Green = A, Red = C. A 10 micron scale bar is included in the bottom right image. **C)** 50th (left) and 99.9th (right) percentile pixel intensity of sequencing images in VIS-seq image set 1.1. The 50th percentile approximates background and the 99.9th percentile approximates the intensity of signal generated by amplicon colonies. **D)** The percent of cells with barcodes found in the previously determined lookup table for each experiment when reads were called by STARCall (green) or the Feldman et al. pipeline (blue). **E)** Test set performance of STARCall (green) and the Feldman et al. pipeline (blue). Five hundred cells were selected and manually annotated with their sequence **F)** The read count per cell of STARCall (left) and the Feldman et al. pipeline (right) on image set 2.1. **G)** The nucleotide frequency at each cycle from reads called by STARCall (left) or the Feldman et al. [4] pipeline (right), with the expected frequency in the barcode library shown by dashed lines.

effect of the increase in background and decrease in signal means that for the C channel in image set 1.1, signal is overwhelmed by background in later cycles (Fig 4C). To ensure accurate base calling, we remove this background by subtracting a Gaussian blur of the image, applied independently for each cycle and channel. This filter removes larger features and general background fluorescence, leaving only the small features characteristic of amplicon colonies (Fig 4B). Next we compute pixel intensity z-scores from each channel and cycle independently, which normalizes the large change in intensity across cycles and between channels (Fig 4C).

To detect the amplicon colonies, we apply a filter that selects for dots that are bright in only one channel. We subtract each pixel's value in the second highest channel from all other channels, resulting in each pixel having only one positive value across channels. We next apply a Gaussian filter with a standard deviation of 1 pixel to reduce any noise introduced by this procedure, then set any negative pixel values to zero. We combine all cycles for each channel by calculating the standard deviation of each pixel's value across cycles. As the amplicon colonies change frequently across cycles, pixels in which colonies are present will have a high standard deviation. We finally sum each pixel's standard deviation value across all channels to create a single grayscale image [4]. Bright dots in this image correspond to amplicon colonies, and we find the locations of all colonies using the Laplacian of Gaussian blob detection algorithm [14].

We next call the base in each cycle as the channel with the maximum pixel intensity at the dot location in the background-filtered image. For each dot, base calls are concatenated to create reads. Lastly, we match reads to cells, segmenting cells with CellPose [11] and assigning each read to the cell within which it is contained. To form consensus reads for each cell, we take advantage of the fact that *in situ* sequencing experiments generally employ a lookup table enumerating the expected sequences, which are barcodes in the case of optical pooled screening. We count each unique read in each cell and retain the two most frequently occurring reads, discarding the rest. If desired, the number of reads retained can be changed by the user based on their experimental design [4]. We search the look up table for the lowest edit distance match to each retained read. If a cell has a single match, we label that cell with the matching barcode. If a cell has multiple matches with the same edit distance, we call the cell ambiguous. Some experiments use double barcodes, in which case matches are made between a pair of barcodes and a pair of reads, with the individual edit distances added together. The final result is that each cell is linked to one or more barcodes that reveal the CRISPR guide or genetic variant introduced into the cell.

To compare the read calling performance of STARCall to the Feldman et al. pipeline [4], we used nine image sets from multiple barcode-based *in situ* sequencing experiments (Table 1). We stitched all sets with ConStitch and used CellPose to generate cell segmentations, then ran the read calling section of both pipelines. We used the barcode lookup tables determined for each experiment, which provided a ground truth for the barcode sequences that should be present in cells.

For the Feldman image set, STARCall yielded 8% more cells with reads matching the lookup table than did the Feldman et al. pipeline (Fig 4D). STARCall yielded substantially more cells with matching barcodes for all eight VIS-seq experiments, particularly for image set 2.1 where 44% of cells had matching barcodes from STARCall whereas only 3% of cells had matching barcodes from the Feldman et al. pipeline. We validated this result using a set of 500 cells from the first 5 image sets with manually annotated sequences (Fig 4E). The performance discrepancy on image set 2.1 arose because STARCall called many more reads per cell than the Feldman et al. pipeline (Fig 4F). We also compared performance by analyzing the relative frequency of base calls in each cycle for image set 1.1 (Fig 4G). The Feldman et al. pipeline yielded a sharp increase in the frequency of C base calls in later cycles whereas STARCall's base calls remained close to the

base call frequencies found in the barcode library lookup table. The rise in C base calls matched the increased background in later cycles, illustrating that STARCall was more robust to the varying levels of background. Thus, STARCall is able to detect more sequencing reads in varying image conditions, providing more reliable and accurate read calls.

**STARCall implementation, data handling and use**

STARCall is implemented in Python, uses Snakemake [16] for organization and is freely available under the permissive MIT license. *In situ* sequencing image sets can comprise tens of terabytes of data, necessitating memory and CPU efficient processing methods. STARCall provides many features that improve performance and allow for the processing of these very large datasets. The main way memory and CPU usage is regulated is by splitting the stitched well images up into smaller tiles, meaning each processing step is not limited by the need to load large stitched images into memory. Additionally, tiles facilitate parallelization, and the size of these tiles can be tuned by the user.

We measured the resource usage for a full run of STARCall on image set 1.1 comprising 5,868 images and totaling 439 Gb. Execution took 19 hours with a peak memory of 854 Gb. The total resource usage was 538 CPU hours and 4.9 Tb hours. The CPU hours used for each step in the pipeline were measured, with stitching and alignment, cell segmentation, dot detection, and feature extraction taking up the majority of the resources (Fig 5A). We did not utilize GPU execution to speed up the cell segmentation step, but the underlying libraries Cellpose and Stardist both support GPU execution. STARCall provides a configuration option to enable GPU execution, which would reduce cell segmentation execution time.

We compared the resources used by ConStitch and ASHLAR [9] when stitching and aligning image set 1.1. ConStitch required less time to run, but used 18.7 CPU hours and had a peak memory usage of 93 Gb compared to only 6.9 CPU hours and 5.9 Gb for ASHLAR (Fig 5B). We also compared the resource usage of the read calling portion of STARCall to the Feldman et al. pipeline [4]. Stitched images and cell segmentation masks were provided to both STARCall and the Feldman et al. pipeline. STARCall required more time to run and used 7.7 CPU hours with a peak memory usage of 99 Gb, compared to 1.8 CPU hours and 109 Gb for the Feldman et al pipeline (Fig 5C). Thus, STARCall delivers superior stitching, alignment and read calling performance at the cost of increased resource usage.

As shown in these tests, the improvements in performance that STARCall provides comes with large resource requirements, larger than alternative methods such as ASHLAR and the Feldman et al. pipeline. These resource requirements make execution on a typical desktop or laptop computer challenging, and all the experiments done here were performed in a cluster environment providing the needed resources. Porting code between cluster environments can be difficult as each cluster is configured differently. However, we implemented STARCall using SnakeMake, which supports many cluster environments including Slurm and common cloud providers. For cluster environments that are not supported by SnakeMake, STARCall centralizes all cluster-specific configuration information in one file that can be easily modified.

## Discussion

STARCall improves stitching, alignment and read calling for barcode-based *in situ* sequencing experiments when compared to previous solutions, including ASHLAR [9] and the Feldman et al. pipeline [4]. STARCall's improved stitching and alignment algorithm provides precise alignment on all image sets we tested, outperforming ASHLAR and MIST. STARCall's novel base calling filters and normalization scheme overcome the loss of signal and increase in background as sequencing proceeds, which are the main issues that complicate the read calling. Finally, STARCall combines these improvements with other methods, CellPose [11] and StarDist [12] for cell segmentation and CellProfiler [13] for feature extraction, to provide a modular, end-to-end pipeline for the processing of *in situ* sequencing data.

Despite the improvements STARCall delivers, it is still imperfect. ConStitch, the STARCall stitching library, has a much higher computational cost compared to ASHLAR. Due to each cycle being aligned to every other cycle, this cost also scales quadratically on the number of cycles. In addition, an important limitation of our read calling method is the z-score normalization of the sequencing images, which assumes that each base will be present in approximately equal

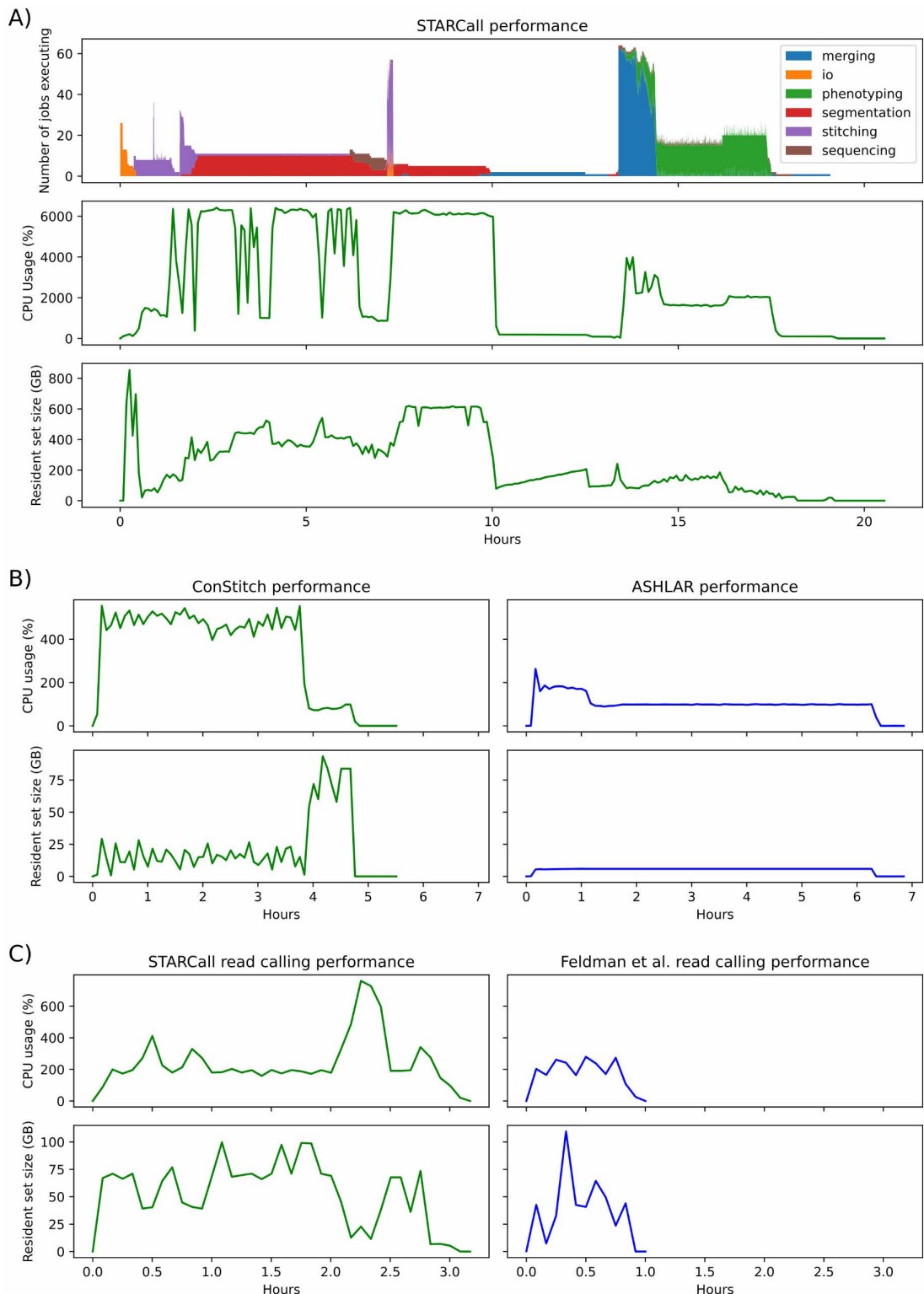

**Fig 5. STARCall resource requirements. A)** Performance metrics for a full run of STARCall on image set 1.1. Shown is the number of jobs running colored by their section of the pipeline (top), the percent of CPU usage measured (middle), and the resident set size (RSS) memory usage in gigabytes (bottom). **B)** Performance metrics for ConStitch, the package that performs stitching and alignment in STARCall, compared to ASHLAR. The percent of CPU usage (top) and RSS memory usage (bottom) over time are shown. **C)** The same performance metrics as in (B) shown for just the read calling portion of STARCall, compared to the Feldman et al. [4] pipeline.

frequencies across reads in every cycle. This may not be the case depending on the experimental design, and a future normalization method could incorporate the actual base frequencies found in the barcode lookup table. Another issue that is common for microscopy images is vignetting; in our tests we did not apply illumination correction to remove the effects of uneven illumination from the images before stitching. These uncorrected images yielded accurate base calls. However, our analysis of subpixel shifting revealed that loss of dot intensity relative to background dramatically increased the mis-calling rate, raising the possibility that severe uneven illumination might degrade read calling performance. We speculate that STARCall is less sensitive to such uneven illumination because it affects the intensity of both dots and background equally, unlike subpixel shifting. However, STARCall can call BaSiC [15] to perform illumination correction if desired.

The algorithms presented in STARCall are aimed at the processing of barcode-based *in situ* sequencing data, but they have a wider range of applications. For example, the stitching algorithms in ConStitch can be used in any imaging experiment where multiple rounds of imaging need to be aligned together, including IBEX [21] and cyclic immunofluores-cence [20], with the possibility of being extended to 3D imaging or time series image sets as well. However these imaging methods pose difficult challenges such as nonlinear sample deformation, so application of STARCall may require large changes. Our approach to identifying amplicon colonies and calling reads can be applied to any barcode-based *in situ* sequencing experiment, and could be extended to adjacent experiment types such as sequencing-based spatial tran-scriptomics. Thus, STARCall is an end-to-end pipeline for analyzing *in situ* sequencing data, and we show that it improves image stitching, alignment and read calling compared to previous approaches.

## Methods

### Implementation and dependencies

STARCall is implemented in Python 3 using the common python libraries NumPy, Scikit-Image, Scikit-Learn, Pandas, tifffile, nd2, PuLP, Numba, and SciPy. CellPose [11] and Stardist [12] are used for cell segmentation, and CellProfiler [13] is used for feature extraction. The software is organized as two Python packages, ConStitch and STARCall, and a SnakeMake pipeline to handle the execution of the different steps in the pipeline. ConStitch contains the algorithm used for stitching and alignment of these multi cycle image sets, and can be applied to various imaging protocols for *in situ* sequencing or otherwise. STARCall contains the remaining steps that are needed for data processing, with improved filters for read detection and extraction, and utilities necessary for cell segmentation, image input/output and conversion, and mapping reads to a barcode library. Both libraries are made to be used separately, however the Snakemake pipeline combines both to process terabytes of image data on a cluster environment.

### Stitching and alignment of images across cycles

When stitching we assume an *in situ* sequencing imaging dataset for one well contains $N_{cycles}$ cycles, each with a set of $N_{tiles}$ images, of dimensions $W \times H$ and $N_{channels}$, taken at different positions across the well plate. Images are taken so that some regions of the well are imaged multiple times, meaning neighboring images have sections that contain the same area, which we call overlap between images. Additional assumptions made for this stitching is that the transformations between images in the same cycle and between cycles are only translational, with no large rotational motion. A scalar transformation can be handled if the level of scaling is specified by the user, so images can be rescaled when alignment occurs. The different channels of each image are also assumed to be aligned to each other, if they were imaged sepa-rately additional measures would be needed to align between them. One of these channels is designated as fiducial chan-nels that will remain constant between different images, any other cycles are ignored during stitching and alignment.

Stitching begins by arranging all images using the positional data provided. If this data is not measured in pixels, for example as positions in a tile grid or in microns, it is converted using the size of images or the pixel scale in microns. Each image $I_i$ is annotated with a position $(x_i, y_i)$, a size $(w_i, h_i)$, and a cycle number $c_i$. If there are images that were

taken at a different scale, then a scale factor is attributed to the image. For example, the phenotyping images for the VIS-seq image sets were taken at 20X while the base images were taken at 10X, so the phenotyping images would have a scale factor of 0.5.

Following initial image placement, we construct a graph with images as nodes and add edges connecting any over-lapping image pair. Each edge between images $i$ and $j$ stores the difference in positions between image $i$ and $j$, defined as $dx_{i,j}$ and $dx_{i,j}$. We will refer to these edges as constraints, as they can be thought of as a positional constraint between the position of two images. To define which image pairs are overlapping we calculate the percent overlap of all possible edges, defined below as $p_{i,j}$. Two types of edges are added, intra-cycle edges which connect neighboring images in the same cycle, and inter-cycle images which connect images in the same position bit in different cycles. These are defined below as the sets $C_{intra}$ and $C_{inter}$, combined to form the set $C$ of all constraints.

$$dx_{i,j} = x_j - x_i$$

$$dy_{i,j} = y_j - y_i$$

$$ox_{i,j} = \max(x_i, x_j) - \min(x_i + w_i, x_j + w_j)$$

$$oy_{i,j} = \max(y_i, y_j) - \min(y_i + h_i, y_j + h_j)$$

$$p_{i,j} = \frac{\max(0, ox_{i,j}) \cdot \max(0, oy_{i,j})}{\min(w_i h_i, w_j h_j)}$$

$$C_{intra} = \{(i,j) : p_{i,j} > 0 \text{ and } c_i = c_j\}$$

$$C_{inter} = \{(i,j) : p_{i,j} \geq 0.5 \text{ and } c_i \neq c_j\}$$

$$C = C_{intra} \cup C_{inter}$$

Using the phase cross correlation algorithm [17,18], every constraint is then refined to its true alignment, calculating an updated $dx'_{i,j}$ and $dy'_{i,j}$ for all $i,j$. If the two images have different scale factors, e.g., they were taken with different objec-tives, then the smaller image is upscaled to match the resolution of the larger one. We use a modified version of phase correlation [10], which begins by calculating the cross correlation using fast fourier transform (FFT). According to the shift theorem, an offset in the time domain corresponds to a linear phase term in the frequency domain. This allows us to cal-culate the cross correlation using only FFT and iFFT operations, greatly reducing the computation necessary. This method calculates the circular cross correlation, meaning a single offset in the resulting image corresponds to four possible offsets of the images. We take the top two maximal positions from the cross correlation matrix and score all 4 offsets for each using the zero normalized cross correlation (ZNCC), defined as the zero normalized dot product of the overlapping sec-tions of the two images, and represented below as the star operator. This gives us 8 possible offsets all scored with the

 

ZNCC, and we take the offset with the maximum score. Constraint values $dx'$ and $dy'$ are updated with the new alignment offset, as well as storing the score for later. These changes are defined below:

$$dx'_{i,j}, dy'_{i,j} = \text{argmax}\left(I_i \star I_j\right)$$

$$s_{i,j} = \left(I_i \star I_j\right)_{dx'_{i,j}, dy'_{i,j}}$$

The next step is to filter out any erroneous constraints, which arise in areas where not many features are shared between images, for example in areas where the plating of cells is sparse. A score threshold is found by finding a set of 100 image pairs that are far from each other ensuring we get an accurate sample of ZNCC scores between images that share no overlap. For each of these pairs $dx'_{i,j}$, $dy'_{i,j}$, and $s_{i,j}$ are calculated as described above and the score threshold is taken as the 95th percentile of all $s_{i,j}$. Any constraint below this threshold is removed from the set of constraints, defined below as $C_{\text{filtered}}$.

$$C_{\text{not overlapping}} = \{(i,j) : dx_{i,j} \leq -2w_i \text{ or } dx_{i,j} \geq 2w_j \text{ or } dy_{i,j} \leq -2h_i \text{ or } dy_{i,j} \geq 2h_j\}$$

$$s_{\text{thresh}} = P_{95}\{s_{i,j} : (i,j) \in C_{\text{not overlapping}}\}$$

$$C_{\text{filtered}} = \{(i,j) : s_{i,j} \geq s_{\text{thresh}}\}$$

A linear model is then trained on the remaining constraint values, with $X = \left[dx_{i,j}, dy_{i,j}\right]$ and $y = \left[dx'_{i,j}, dy'_{i,j}\right]$ for all $i,j$. Additionally an outlier resistant regression method such as RANSAC [22] is used to remove any additional erroneous constraints that still remain after filtering. This linear model is used to replace any constraints that were removed by the score threshold or the outlier classification, calculating replacement values as defined below, where $\hat{dx}$ and $\hat{dy}$ are the fitted linear model for $dx'$ and $dy'$.

$$dx'_{i,j} = \hat{dx}_{i,j} \text{ where } (i,j) \in C \text{ and } (i,j) \notin C_{\text{filtered}}$$

$$dy'_{i,j} = \hat{dy}_{i,j} \text{ where } (i,j) \in C \text{ and } (i,j) \notin C_{\text{filtered}}$$

The final step is to find global positions for all images, using the refined offsets $dx'$ and $dy'$. To do this we can use the original equations used to calculate $dx$ and $dy$ and set up a system of equations for $x'_i$ and $y'_i$ for all images. Each constraint adds two equations to the system, detailed below. We can also incorporate the scores of constraints by multiplying them to both sides of the equation, which results in error being minimized more for constraints with a high score.

$$s_{i,j}\left(x'_j - x'_i\right) = s_{i,j}\left(dx'_{i,j}\right)$$

$$s_{i,j}\left(y'_j - y'_i\right) = s_{i,j}\left(dy'_{i,j}\right)$$

By solving this system of equations of all $x'_i$ and $y'_i$, we obtain global positions for all images. Because of the large number of constraints, this system will be overconstrained, and a perfect solution is not possible. We can use any linear regression solver to find the solution to this system. Various loss functions can be minimized, but we use the mean

absolute error. To find this solution, we construct a linear programming problem. We introduce slack variables $t_{i,j}$, $u_{i,j}$, $v_{i,j}$, $w_{i,j}$ into the equations above, and minimize the sum of all slack variables. With integer linear programming the values of $x'_i$ and $y'_i$ can also be constrained to only integers. Using a linear programming solver, currently the PuLP Python library, with the problem specified below, we find the values for all $x'_i$ and $y'_i$.

$$
\begin{aligned}
\text{minimize} \quad & \sum_{i,j \in C} \left( t_{i,j} + u_{i,j} + v_{i,j} + w_{i,j} \right) \\
\text{subject to} \quad & s_{i,j}\left( x'_j - x'_i \right) = s_{i,j}\left( dx'_{i,j} \right) + t_{i,j} - u_{i,j} & (i,j) \in C \\
& s_{i,j}\left( y'_j - y'_i \right) = s_{i,j}\left( dy'_{i,j} \right) + v_{i,j} - w_{i,j} & (i,j) \in C \\
& t_{i,j} \geq 0 & (i,j) \in C \\
& u_{i,j} \geq 0 & (i,j) \in C \\
& v_{i,j} \geq 0 & (i,j) \in C \\
& w_{i,j} \geq 0 & (i,j) \in C \\
& x'_i \in \mathbb{Z} & i = 1 \ldots N_{\text{tiles}} \\
& y'_i \in \mathbb{Z} & i = 1 \ldots N_{\text{tiles}}
\end{aligned}
$$

With global positions for each image, we can combine all images in each cycle together, merging overlapping areas by either taking the mean of all values, or by taking the pixel intensity that is closest to the center of its respective image. All cycles are combined in this manner to create a full stitched image for every cycle, with each cycle aligned to one another.

## Evaluation of alignment

To evaluate the alignment of the stitched images across cycles, we split each cycle up into a grid of tiles 200 pixels by 200 pixels. At each tile, we applied the phase cross correlation algorithm [17,18] to ensure the alignment of the small tile is optimal, calculating the alignment between every pair of cycles. If alignment is optimal we expect these alignments to all be zero, and we calculated the error as the L2 norm of these alignments. On image sets with more than two cycles, we calculate the error between all cycle pairs, and take the maximum alignment error.

## Execution of ASHLAR

To obtain stitched images from ASHLAR, we used the commands in S1 Table. Because we wanted to test ASHLAR with and without subpixel interpolation, we modified the source code to add a parameter "--no-resample" that disabled interpolation when stitching. We also added a parameter "--transpose" that swaps the x and y axes of the microscope stage positions, which was required to stitch some of the image sets. These changes are available at (https://github.com/njbradley/ashlar).

## Read calling

When detecting reads, we assume amplicon colonies will present as small, high frequency features that are bright in only one channel and change frequently across cycles. This contrasts with the different sources of noise we want to filter out, such as cells which are larger, lower frequency features that are dim in all channels and increase slowly across cycles. Cell debris is another source of noise, which are smaller, higher frequency features bright in all channels and not changing greatly across cycles.

We begin with a series of image filters, as input we take all cycles and all channels, meaning we have a 4 dimensional image with dimensions (num_cycles, num_channels, width, height). First we subtract a small (sigma = 3) Gaussian blur from each image, applying this to every cycle and channel separately. This reduces the intensity of any features larger

than the sigma. By using a sigma larger than our features of interest, this greatly reduces the intensity of larger features such as cells. In addition, the cell background is subtracted from the reads contained within it, which will be important when extracting read values. We then z-score the pixel intensities of each channel and cycle independently. To further highlight the reads, we subtract the second-maximum value across channels on a per-pixel basis.

Because the reads in the image are only bright in one channel, the second maximal channel will be low and they will be preserved through this filter. On the other hand features that are bright in all channels, such as cells or cell debris will be greatly reduced. This operation can amplify the background noise present, and a small Gaussian blur (sigma = 1) is applied to the resulting image to remove such noise. Clipping any negative values to zero leaves us with only features bright in one channel. The final processing step combines all images across cycles and channels, first across cycles by taking the standard deviation across cycles on a per-pixel basis, then summing across channels.

Because the sequencing reads change every cycle, when applying the standard deviation across cycles we expect a large resulting number. However, features that do not change much, such as debris or cell background, will not result in a large standard deviation across cycles, and will be filtered out.

With this resulting 2 dimensional image, we utilize the Laplacian of Gaussian blob detection algorithm [14] to identify small blobs in the grayscale image. We use the implementation provided in the scikit-image python library, with parameters min_sigma = 1, max_sigma = 3, num_sigma = 7. The resulting blobs detected are our sequencing reads, and we use these positions to extract the read values, returning to our original image and indexing at said positions. For each sequencing read we have values for 12 cycles by 4 channels, with the sequence encoded inside. To call this sequence we take the maximum channel for each cycle, resulting in a 12 base pair sequence for each peak detected.

In order to match reads to cells we first need to identify and segment the cells present, for which we use the deep learning model CellPose. CellPose provides a general and robust method for cell segmentation, and we found it provided high quality cell segmentation on our input images.

Using the cell segmentation mask from Cellpose we correspond each sequencing colony to a cell. To select consensus reads, we take the top two most frequent reads present in each cell, and discard the rest. We then match each cell to a library barcode using the edit distance between each read corresponding to the cell and each library barcode. Cells which contain a perfect match for a library barcode can be linked easily, otherwise cells are only linked if there is a single library barcode with a minimal edit distance to one of the reads in the cell. If multiple library barcodes are the same edit distance away, the cell is unable to be matched. For image sets VIS-seq 3 and 4, the library is constructed where each cell contains two barcodes, and thus the library is a set of double barcodes. Cells are matched to barcode pairs using the combined edit distance of both barcodes, with the same logic as before.

## Read calling test set

The test set for read calling was created by manually annotating the sequences of 500 cells from the first five image sets, 100 from each set. This was done using an interactive web server, which presented crops of the cell for each cycle. Cells were selected randomly, and any cells that we were unable to confidently annotate were removed and replaced with a new random cell. This was repeated until we had 100 cells from each image set. The sequences were verified with the barcode library for each image set, and errors were corrected. Test set accuracy was measured by comparing the reads provided by the read calling method to the expected sequences, if the cell contained the expected sequence it was counted as accurate. No partial credit was given to sequences that differed by a single base, only exact matches were counted.

## Cell segmentation

Cell segmentation is performed for cells and cell nuclei, using Stardist [12] for nuclear segmentation and CellPose [11] for cell segmentation. CellPose provides a general and robust method for cell segmentation, which takes as input a channel

for the nucleus and a channel for the cytoplasm. On the VIS-seq images we used DAPI as the nuclear channel, and for image sets 1 and 2 a WGA+Phalloidin combination channel for the cytoplasm, and for image sets 3 and 4 a Phalloidin only channel. CellPose also requires a diameter parameter, for which we used 100 pixels. This was found through manual inspection of the images, and resulting segmentation was inspected for each image set to ensure performance was adequate

To segment the *in situ* sequencing image set in Feldman et al. [4] we used DAPI as the nuclear channel, and the C sequencing channel from the final sequencing cycle as the cytoplasm channel, as the C dye builds up more than the other dyes and provides a functional cytoplasm channel. For the diameter parameter we used 50 pixels.

## Phenotyping

We measure a cell's phenotype using CellProfiler [13], a proven method for extracting a large set of features that represent the phenotypic state of a cell. When running the pipeline, a CellProfiler pipeline file can be provided that will be run on the stitched phenotype images and the cell segmentation. As input the cellprofiler pipeline receives all phenotype channels labeled CH0 to CHN, and the cell and nuclear segmentations named Cells and Nuclei. Any of the diverse CellProfiler modules can be used in the pipeline to calculate different features on cells, nuclei, or custom areas of the cell. At the end of the pipeline, all important features are matched to the original cell segmentation, and features are exported to Cells. csv from the pipeline. This file is then concatenated to the other output files including sequencing information, providing a phenotype to genotype output file.

## Resource usage

We measured the resource usage of STARCall on a cluster node with 64 cores and 1T of RAM available. When comparing the resource usage of ConStitch and ASHLAR, both were run on the same cluster node with 8 cores and 100GB of RAM available. The same cluster environment was used to compare the resource usage of STARCall's read calling and the Feldman et al. pipeline [4]. Memory and CPU usage were measured every 10 seconds during execution.

## Supporting information

**S1 Table. Commands used to invoke ASHLAR.**
(CSV)

## Acknowledgments

We would like to acknowledge William Noble and Gwenneth Straub for computational assistance and testing. For maintaining the cluster used in this analysis we thank the Genome Sciences Information Technology Services Team.

## Author contributions

**Conceptualization:** Nicholas J. Bradley.

**Data curation:** Nicholas J. Bradley.

**Formal analysis:** Nicholas J. Bradley.

**Funding acquisition:** Douglas M Fowler.

**Investigation:** Nicholas J. Bradley, Sriram Pendyala, Katie Partington.

**Methodology:** Nicholas J. Bradley.

**Project administration:** Douglas M. Fowler.

**Resources:** Nicholas J. Bradley.

**Software:** Nicholas J. Bradley.

**Supervision:** Sriram Pendyala, Douglas M. Fowler.

**Validation:** Nicholas J. Bradley.

**Visualization:** Nicholas J. Bradley.

**Writing – original draft:** Nicholas J. Bradley.

**Writing – review & editing:** Nicholas J. Bradley, Sriram Pendyala, Katie Partington, Douglas M. Fowler.

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
