## [Decision Letter · Decision Letter 0]

13 Jan 2026

PCOMPBIOL-D-25-02250

STARCall integrates image stitching, alignment, and read calling to enable scalable analysis of in situ sequencing data

PLOS Computational Biology

Dear Dr. Fowler,

Thank you for submitting your manuscript to PLOS Computational Biology. After careful consideration, we feel that it has merit but does not fully meet PLOS Computational Biology's publication criteria as it currently stands. Therefore, we invite you to submit a revised version of the manuscript that addresses the points raised during the review process.

We look forward to receiving your revised manuscript.

Kind regards,

Robert F. Murphy

Guest Editor

PLOS Computational Biology

Jean Fan

Section Editor

PLOS Computational Biology

**Additional Editor Comments:**

Thank you for submitting your interesting manuscript. As you can see, the reviewers all viewed it very positively. However, they had a number of questions and comments. While I don't think each of these is particularly major, the large number of them suggests that some significant revisions will be needed to make the manuscript suitable for publication. In your revision, please carefully describe how you have addressed each concern. Note that it is not necessary to follow every suggestion, but it is necessary to explain your reasoning. We look forward to receiving your revision!

**Journal Requirements:**

At this stage, the following Authors/Authors require contributions: Nicholas J. Bradley, and Katie Partington. Please ensure that the full contributions of each author are acknowledged in the "Add/Edit/Remove Authors" section of our submission form.

4) Please amend your detailed Financial Disclosure statement. This is published with the article. It must therefore be completed in full sentences and contain the exact wording you wish to be published.

State what role the funders took in the study. If the funders had no role in your study, please state: "The funders had no role in study design, data collection and analysis, decision to publish, or preparation of the manuscript.".

**Reviewers' comments:**

Reviewer's Responses to Questions

**Comments to the Authors:**

Reviewer #1: The authors present a pipeline for performing read calling of in situ sequencing starting from image sequences of the fluorescently labeled nucleotides. The pipeline’s novel algorithms for image alignment algorithm and spot-to-base calling build on existing state-of-the-art methods, claiming improved accuracy and robustness. Image alignment and base calling performance is evaluated on a number of datasets, some provided by the authors and some from other publications. The performance evaluation supports the claim that this pipeline is a marked improvement over existing methods, and the authors identify known weaknesses and areas of potential improvement. An end-to-end snakemake pipeline is provided for ease of use as well as standalone python packages for flexible integration with other analyses. Overall this work is a novel and valuable contribution to the field.

I would recommend this manuscript for publication with minor changes and clarifications as follows:

1. There is no mention of flat field image correction in the pipeline. All microscopes suffer from vignetting to some degree, causing measured fluorescence intensity to drop off by 50% or more at the edges of an image. Given that the text claims a 10% drop in spot intensity due to subpixel image shifting leads to a read miscall rate of 15%, it seems like a 50% drop due to vignetting would be catastrophic. Is your microscope software already applying illumination correction? Or perhaps the base calling algorithm is somehow insensitive to vignetting yet highly sensitive to the subpixel shift issue? This needs to be explained in the text.

2. Figure 2 f,g,h: The error metric values for MIST and ASHLAR seem quite different from what was reported in their respective papers (even after correcting for different units – microns vs. pixels – in the ASHLAR case). Specifically, I am looking at MIST Table 1 and Ashlar Figure 5A/5B. Can you explain these discrepancies?

3. Figures overall: A lot of panels are missing appropriate units on some of the axes. Images lack scale bars. (In figures containing multiple same-scaled image fields, you can add the scale bar to just one convenient field)

1A: scale bar

2B: scale bar

2C: Y axis - what is the unit for alignment error?

2D: upper panel Y axis: What is the dot intensity unit? (A.U. is OK, but say so)

2E: X axis

2H: X axis

2J: X axis

3C: Y axis (both panels)

4. Lines 61, 110, 194: The text does not explain how alignment of images from different objectives or microscopes are handled. Different objectives or microscopes will have different magnification and/or sensor pixel pitch, leading to images with different nominal image resolution (pixel size) which phase correlation alone cannot handle. The manuscript must explain how this is addressed in the algorithm.

5. Line 222: Gaussian filters can be parameterized in multiple ways, and describing it as “1 pixel” is not precise enough. I assume the authors meant “a Gaussian filter with a standard deviation of 1 pixel”. Please clarify in the text.

6. Line 289: Here the stitching alignment is claimed to provide “subpixel alignment,” but the algorithm forces image positions to integer pixel coordinates. Please explain this discrepancy.

7. Remember to provide all data used in creating the various figure plots, as per the PLOS Data Availability Policy: “For example, authors should submit the following data: The values behind the means, standard deviations and other measures reported; The values used to build graphs; …” The preferred approach is submitting the data files to an accepted public data repository.

Reviewer #2: Specific questions/comments:

1. “This resulted in more than 50% of tiles with <1 pixel alignment error on all nine image sets, while ASHLAR had higher error on four image sets” - How big was the difference between the image sets? Is there a pattern in images where ASHLAR and STARCall have similar quality and when they differ?

2. Fig 2b – ‘Filter out regions without features’ – what does filter out mean? What if there is useful information in those regions that gets filtered out?

3. Fig 2c. caption ‘Gaussian noise of sigma=100 was added’ – what is the sigma for the image prior to adding gaussian noise? What dtype is used?

4. Fig 2f. median percent error of ROI area is higher for Constitch as opposed to MIST – is this an acceptable trade-off?

5. In the text figure 2b step 2, etc are mentioned. It would be good to add an annotation withing the figures itself for step number for clarity.

6. “A linear model is trained on the remaining constraints to estimate physical parameters of the microscope such as the travel between tiles and the camera angle” – this is interesting – how well does a linear model fare and are there similarities observed between microscopes?

7. MAE solver is likely more robust because it is less affected by outliers as compared to MSE, which penalizes large deviations heavily.

8. The result from interpolation vs 1 pixel alignment error is interesting. What type of interpolation was used?

9. Removing unwanted high frequency features such as cell debris using subtraction method is simple and effective. However, if the high frequency feature has different intensity across images they might still disrupt the algorithms, even if the subtracted intensity is low. How likely are such variations to occur? Is the standard deviation always low in such cases?

10. STARCall CPU time is significantly higher than Feldman et al. Are there any plans for leveraging GPUs for speeding up computation?

11. How was the diameter parameter for Cellpose segmentation chosen?

Reviewer #3: This paper presents STARCall, a computational pipeline for stitching, cycle-to-cycle registration, and read calling of in situ sequencing image datasets which additionally interfaces with existing open source cell segmentation and cell image phenotyping packages (Cellpose, Stardist, and CellProfiler).

*** Summary ***

--- stitching ----

(A1) Relative translations for all tile overlap regions are estimated using Fourier Phase Correlation (FPC).

(A2) Overlaps with insufficient image features for registration are automatically detected and removed if their post FPC score is below the 95th percentile of scores for FPC alignments of randomly selected non-overlapping regions.

(A3) A linear model is fit to the retained translations which provides translations for overlaps removed in the previous step; essentially, missing translations are linearly interpolated from known translations.

(A4) Each tile has an optimal translation with respect to each of its neighbors - which do we accept? This over constrained system is solved for single optimal positions for each tile with an integer linear program (ILP) that minimizes the sum of "slack variables." The "slack variables" are defined as the absolute error between a tiles position and the neighbor-optimal translation found with FPC.

--- cycle-to-cycle alignment ---

(B1) This crucial step is not thoroughly described anywhere in the main text or methods section. This is an essential missing detail that should be described in explicit detail before this paper is considered for publication.

--- read calling ---

Aligned data can be considered a 4-dimensional array: f(t, c, x, y) for cycles, channels, width, and height.

(C1) For all t and c, a Gaussian smoothing of sigma=3 pixels defines a low frequency background which is subtracted from the original images.

(C2) To isolate signal amplitude of true positive labels, the second maximum over c is subtracted in a pixel wise manner at all locations x and y and for all cycles t.

(C3) For all t and c, a Gaussian smoothing of sigma=1 pixels is applied to remove high-frequency noise.

(C4) To highlight amplicon colonies, the standard deviation is taken over t then a sum is taken over c.

(C5) A Laplacian of Gaussian (LoG) filter detects local maximums in the 2D image from the previous step.

(C6) A base pair read is called at every LoG detected point as the maximum value over c at that spatial location.

--- experiments ---

(D1) A few implementation details are compared: the ILP solver and interpolation method used when spatially shifting tiles/cycles.

(D2) The MIST dataset includes evaluation images wherein cell colonies are carefully centered in the single tile acquisition field of view. A second acquisition of the same sample is arbitrarily tiled. Ideally, after the second dataset is stitched, cell colony positions and areas (obtained by post-stitching segmentation) should not significantly differ from the evaluation images.

(D3) This paper asserts that after stitching and cycle-to-cycle alignment any residual misalignment in image sub-regions must be induced by stitching error (rather than true sample deformation between cycles). 200x200 pixel sub-regions from stitched images are registered with FPC between cycles; the measured shift is considered stitching error. When the authors frequently state "less than 1 pixel alignment error" it is with respect to this indirect measure that they are referring.

(D4) The percent of segmented cells which are confidently genotyped after read calling is provided as a measure of success with no effort to quantify false positive or false negative rates.

(D5) Given that the target sequences are known, the frequency of bases in the library sequences should approximately match the frequency of bases called by the read method. This comparison is also provided.

--- implementation, software, and performance ---

(E1) All software is open source and at first shallow glance the repositories seem well documented and well organized.

(E2) Run times using a 100 CPU + 1TB RAM cluster job, as quantified by the cluster manager itself (from the appearance of the graphs) are presented.

*** Comments ***

--- stitching ---

Translation only stitching using FPC is a very well established method, nothing in steps A1, A2, or A3 appears novel to me. Regarding step A4, given the slack variables definition, it seems to me that after accounting for cancelations and doing some algebra the ILP objective function reduces to exactly the cumulative error between tile positions and the overlap constraints and is thus only trivially different from what is used in reference 18 by using MAE instead of MSE. Scaling the constraint equations by the overlap FPC scores seems novel, but also dubious. For realistic image data, in my experience correlation is a useful relative measure, but not a useful absolute one. The correlation of a well aligned overlap region can be significantly changed by adding noise or removing/adding a few cell colonies. So, the FPC score is as much a measurement of the content contained in the overlap region as it is of the alignment quality. Overall I would say the method used here is a standard and well established one with a few tweaks cast to an ILP formulation.

--- cycle-to-cycle alignment ---

A thorough description of this is not apparent in the text anywhere that I can find. The methods section on stitching and alignment should be expanded to discuss alignment across the cycle axis in as much detail as alignment across tiles is described; especially because the authors claim that doing stitching and cycle-to-cycle alignment jointly improves upon prior approaches. If the authors want to keep this claim in the paper, it should be directly supported by experiments that compare a joint optimization to separate optimization of tile and cycle alignments.

The most basic assumption is that after stitching FPC is again used to find a single global translation from each cycle to a single target cycle. I have never worked with 2D in situ sequencing data, but in my experience with 3D in situ transcriptomics, between cycle registration has always required more than a single global translation. In fact, for the most sensitive experiments, non-linear deformation between cycles has been required. This is highly sample preparation dependent of course, so I'm not doubtful that the authors were careful and accurate in their own experiments, but the idea that this assumed approach could generalize to 3D or more complex samples is very naive.

--- read calling ---

Similar to the stitching approach, none of the steps used here on their own seem novel to me. However, I do think they are good choices arranged in a good order and they are similar to other published approaches for detecting individual bright spots close in size to the image sampling rate (1 - 3 pixels across).

I noticed the main text and methods sections describing read calling are not entirely consistent with each other. For example, the main text says a difference of Gaussian filter is used, and the methods section does show both low and high pass filters, but with a critical step in between them, which is not exactly a difference of Gaussians filter. Maybe more importantly, the main text describes z-scoring the channels but the methods section does not. These inconsistencies should be ironed out. The method which was actually used to process data presented in the figures should be described and nothing else.

About the z-scoring, it does make sense to do this because the channels are arithmetically combined (a subtraction first, then a sum later). But the authors also make a good point that this assumes similar distributions of bright dots vs. background in all channels. Just a suggestion, an alternative approach would be to do a weak initial spot detection (e.g. just run LoG directly with a low num_sigma), then take the average of the, say, 100 brightest spots and 100 darkest spots and linearly transform the image such that these two numbers equal standard values consistent between channels.

--- experiments ---

Experiments D1, D2, and D5 all seem fine to me, in particular I find it very interesting that interpolation affected spot reading far more than small spatial shifts.

Personally, I do not think experiment D3 is sufficient to make the claim "less than 1 pixel alignment accuracy." In the image registration literature that kind of claim is usually made in reference to gold standard correspondences, e.g. manual annotations or sparse sets of cells expressing a distinct marker not used for registration itself (an independent validation channel). Experiment D3 uses residual misalignment of sub-regions between cycles instead, which is pretty indirect and confounded by the possibility of real sample deformation.

I also do not particularly like the readout for experiments D4, as it's not really a measure of correctly read cells, just the total number of read cells. One could get 100% on this measure by just assigning a random sequence from the library to every cell. I think taking the time to manually inspect some of the amplicon colonies across cycles/channels and having a manually annotated gold standard set, even for just 100 cells say, would be a more trustworthy measure.

--- implementation, software, and performance ---

I did not install or run the software myself, but the repositories look welcoming to me. The paper talks about requiring cluster resources. From personal experience every cluster is quite different and moving code from one cluster to another can often require source code changes. Information in the paper on which cluster managers are already supported and how much effort might be required to move the code to a different cluster manager (from slurm to sge for example) would be very helpful.

I noticed that the cellpose segmentation step took a very long time, something like 8 hours. The paper never mentions a GPU of any kind, so I assume cellpose was run serially on blocks using all cpus to process one block at a time. This is probably not the most efficient way to do this. First, using a GPU is orders of magnitude faster, and second, running multiple blocks in parallel would probably utilize the resources more completely (multiple GPUs would be the best if you have access). I believe cellpose has a distributed version which can handle running blocks in parallel, google "distributed cellpose big data" and you should find it.

*** General comments ***

While I don't think any of the individual steps in this pipeline are by themselves novel, and some descriptions are just missing (cycle-to-cycle alignment), I think the composition of these functions into a single software package targeting an important data type is a worthy accomplishment that should be published. The language in the paper should be temperate about algorithmic novelty and also about criticizing the capability of existing solutions. Regarding that second point, the claim that images taken with different objectives or from different microscopes just can't be registered is made several times in the paper. This is just not at all true. The image registration community has been registering data taken on different instruments for 50 years. Mutual information exists as a registration optimization function for this very reason. Truly novel deep learning methods are being formulated to address this problem as well (e.g. google: Guiding Registration with Emergent Similarity from Pre-Trained Diffusion Models).

Finally, the most important things for me to see this paper get published are:

* Complete description of cycle-to-cycle alignment in main text and methods

* Soften claims about algorithmic novelty, the novelty is in the pipeline, not really the individual steps

* Soften claims about performance when the evaluation method doesn't really support it (experiments D3 and D4)

* A few more comments about what environments are supported by the packages by default (e.g. cluster manager, operating system) and how much coding would be required to run them in a different environment

Things that would be nice:

* pseudocode descriptions of the pipeline steps for quick reference

Oh and one final suggestion:

At the very end of the pipeline, if a read sequence has more than one match in the library with equal Hamming distance, you discard that read as ambiguous. If, during the read calling, instead of calling each base as the single maximum over all channels, you just assign a probability to each of the four bases based on relative signal across the channels, and you carry this through to the library look up step, you can probably resolve these ambiguous cases. What I mean is, not all base substitution errors are equal if you reference back to the image data. Suppose you have a read sequence with two matches in the library with a Hamming distance of 1. Maybe in one case it's A  G and in the other case it's an A  C. Maybe one of these errors is worse than the other with respect to the original image data. If you retain probabilities for each base at each position you can work this out at library look up time. But if you just discard that information and keep the maximum, these errors look the same.

**Have the authors made all data and (if applicable) computational code underlying the findings in their manuscript fully available?**

The PLOS Data policy requires authors to make all data and code underlying the findings described in their manuscript fully available without restriction, with rare exception (please refer to the Data Availability Statement in the manuscript PDF file). The data and code should be provided as part of the manuscript or its supporting information, or deposited to a public repository. For example, in addition to summary statistics, the data points behind means, medians and variance measures should be available. If there are restrictions on publicly sharing data or code —e.g. participant privacy or use of data from a third party—those must be specified.requires authors to make all data and code underlying the findings described in their manuscript fully available without restriction, with rare exception (please refer to the Data Availability Statement in the manuscript PDF file). The data and code should be provided as part of the manuscript or its supporting information, or deposited to a public repository. For example, in addition to summary statistics, the data points behind means, medians and variance measures should be available. If there are restrictions on publicly sharing data or code —e.g. participant privacy or use of data from a third party—those must be specified.

Reviewer #1: **No:** Data points behind graphs are not provided. Code and example input data for that code IS provided, though.Data points behind graphs are not provided. Code and example input data for that code IS provided, though.

Reviewer #2: Yes

Reviewer #3: Yes

PLOS authors have the option to publish the peer review history of their article (what does this mean?). If published, this will include your full peer review and any attached files.). If published, this will include your full peer review and any attached files.

.

Reviewer #1: No

Reviewer #2: **Yes:** Pushkar Sunil SathePushkar Sunil Sathe

Reviewer #3: No

**Figure resubmission:**
---

## [Editor Report · Decision Letter 1]

10 Apr 2026

Dear Dr. Fowler,

We are pleased to inform you that your manuscript 'STARCall integrates image stitching, alignment, and read calling to enable scalable analysis of *in situ* sequencing data' has been provisionally accepted for publication in PLOS Computational Biology.sequencing data' has been provisionally accepted for publication in PLOS Computational Biology.

Best regards,

Robert F. Murphy

Guest Editor

PLOS Computational Biology

Jean Fan

Section Editor

PLOS Computational Biology

Thanks for your careful revisions in response to the initial reviews.

---

## [Editor Report · Acceptance letter]

PCOMPBIOL-D-25-02250R1

STARCall integrates image stitching, alignment, and read calling to enable scalable analysis of *in situ* sequencing datasequencing data

Dear Dr Fowler,

I am pleased to inform you that your manuscript has been formally accepted for publication in PLOS Computational Biology. Your manuscript is now with our production department and you will be notified of the publication date in due course.

With kind regards,

Lilla Horvath
